# OptimAI: Optimization from Natural Language Using LLM-Powered AI Agents

## Abstract

Optimization plays a vital role in scientific research and practical applications. However, formulating a concrete optimization problem described in natural language into a mathematical form and selecting a suitable solver to solve the problem requires substantial domain expertise. We introduce **OptimAI**, a framework for solving Optimization problems described in natural language by leveraging LLM-powered AI agents, and achieve superior performance over current state-of-the-art methods. Our framework is built upon the following key roles: (1) a *formulator* that translates natural language problem descriptions into mathematical formulations; (2) a *planner* that constructs a high-level solution strategy prior to execution; and (3) a *coder* and a *code critic* capable of interacting with the environment and reflecting to refine future actions. Ablation studies confirm that all roles are essential; removing the planner or code critic results in $5.8\times$ and $3.1\times$ drops in productivity, respectively. Furthermore, we introduce UCB-based debug scheduling to dynamically switch between alternative plans, yielding an additional $3.3\times$ productivity gain. Our design emphasizes multi-agent collaboration, and our experiments confirm that combining diverse models leads to performance gains. Our approach attains 88.1% accuracy on the NLP4LP dataset and 82.3% on the Optibench dataset, reducing error rates by 58% and 52%, respectively, over prior best results.

## 1 Introduction

Optimization plays a foundational role across a broad spectrum of scientific and engineering disciplines, serving as a core framework for decision-making, resource allocation, system design, and beyond Gill et al. (2019). In recent years, its importance has grown even more pronounced with the rise of data-driven methodologies. Modern machine learning, in particular, is fundamentally built on solving large-scale optimization problems. Training a model typically involves minimizing a loss function, often over high-dimensional, nonconvex landscapes, and tasks such as hyperparameter tuning, model selection, and policy learning in reinforcement learning are likewise formalized as optimization problems Boyd (2004); Bottou et al. (2018).

Despite its foundational role, optimization remains largely inaccessible to non-experts because real-world objectives are rarely expressed mathematically. Translating goals like minimizing delivery time or balancing risk into formal models demands substantial expertise in both problem formulation and solver selection, which significantly impacts efficiency and feasibility. Recent progress in natural language processing and symbolic reasoning opens the door to automated systems that bridge this gap. By interpreting intuitive problem descriptions and constructing valid optimization formulations, such systems can significantly lower the entry barrier and expand access to powerful optimization technologies.

Large Language Models (LLMs) have emerged as powerful tools for interacting with complex tasks through natural language, offering a user-friendly interface and significant computational capabilities. This work aims to reduce the barrier of translating real-world problems into formal mathematical models, support users in solving these problems, and investigate the extent to which LLMs can reason about optimization. Recent studies have demonstrated

that LLMs exhibit a nontrivial degree of math, coding, and logical Guan et al. (2025); Li et al. (2025a); Zhang et al. (2024b); Achiam et al. (2023) reasoning abilities. Our study contributes to this growing body of research by evaluating the reasoning capabilities of LLMs in the context of optimization tasks.

We introduce OptimAI, a framework for solving optimization problems described in natural language by leveraging LLM-powered AI agents. OptimAI comprises four stages (Figure 1), each handled by a specialized agent. OptimAI offers several key advantages. First, it adopts a **plan-before-code** strategy, generating multiple solution plans before code generation. Second, we introduce **UCB-based debug scheduling** to dynamically switch between alternative plans during debugging, enabling adaptive plan selection based on observed feedback. Practically, distinct mathematical formulations and solver backends can yield large gaps in solution quality and compute cost; exploring alternative models often shrinks problem size substantially, which benefits non-experts who lack domain heuristics. Third, it naturally supports **multi-agent collaboration**, allowing different roles to be handled by distinct LLMs best suited for each task. In our experiments, we observed that combining different LLMs can yield synergistic effects. These design choices contribute to OptimAI's superior performance over prior methods. Table 1 compares the functional capabilities of OptimAI and previous methods.

Comprehensive experiments show that our approach consistently outperforms state-of-the-art methods on several benchmarks. Specifically, compared to the previous best approach, we reduce the error rate on NLP4LP by 58%, and on the four Optibench subsets (Linear w/o Table, Linear w/ Table, Nonlinear w/o Table, and Nonlinear w/ Table) by 48%, 47%, 68%, and 41%, respectively. Ablation studies confirm that all roles are essential; removing the planner or code critic results in 5.8× and 3.1× drops in productivity, respectively. Furthermore, UCB-based debug scheduling yields an additional 3.3× productivity gain. Moreover, OptimAI is broadly applicable. Beyond standard mathematical programming, it also handles NP-hard combinatorial optimization problems, demonstrating strong generality.

Table 1: Comparison of Functional Capabilities between OptimAI and Prior Methods.

| Functional Capabilities | OptiMUS | Optibench | CoE | OptimAI |
|---|---|---|---|---|
| Natural language input | ✗ | ✓ | ✓ | ✓ |
| Planning before coding | ✗ | ✗ | ✓ | ✓ |
| Multi-solver support | ✗ | ✗ | ✗ | ✓ |
| Switching between plans | ✗ | ✗ | ✗ | ✓ |
| Code generation | ✓ | ✓ | ✓ | ✓ |
| Distinct LLM collaboration | ✗ | ✗ | ✗ | ✓ |

## 2 RELATED WORK

The application of LLMs to complex computational tasks has received growing attention in recent years. Our work lies at the intersection of three emerging research directions: leveraging LLMs for solving optimization problems, enhancing their reasoning capabilities, and enabling multi-agent collaboration for coordinated problem solving. In this section, we review recent advances in each of these areas, which collectively motivate our integration of these components into a unified framework for solving optimization tasks.

Table 2: Previous work on using LLMs for optimization.

| Work | Dataset Proposed | Size | Problem Type(s) |
|---|---|---|---|
| NL4Opt Competition Ramamonjison et al. (2023) | NL4Opt | 289 | LP |
| Chain-of-Experts (CoE) Xiao et al. (2023) | ComplexOR | 37 | LP, MILP |
| OptiMUS AhmadiTeshnizi et al. (2023; 2024; 2025) | NLP4LP | 67 | LP, MILP |
| Optibench Yang et al. (2024) | Optibench | 605 | LP, NLP, MILP, MINLP |
| OR-LLM-Agent Zhang & Luo (2025) | OR-LLM-Agent | 83 | LP, MILP |

Abbreviations: LP - Linear Programming, NLP - Nonlinear Programming, MI - Mixed-Integer.

## 2.1 LLM for Optimization

Recent work has explored the use of LLMs to model, interpret, and solve optimization problems directly from natural language descriptions, enabling new interfaces between human intent and mathematical problem solving. Approaches range from prompting LLMs with structured problem templates to integrating them with external solvers for tasks such as (mixed-integer) linear and nonlinear programming problems. Table 2 summarizes existing works, including the datasets they propose, the size of each dataset, and the types of optimization problems they support.

NL4Opt Ramamonjison et al. (2023) introduced the natural language for optimization competition, which comprised two sub-tasks: (1) recognition of optimization problem entities and (2) generation of math formulations. Notably, GPT-3.5 outperformed the competition winner in both tasks. However, ChatGPT exhibited common errors such as incorrect constraint coefficients, redundant constraints, and omitted variables. These findings highlight both the potential and limitations of LLMs in optimization.

OptiMUS AhmadiTeshnizi et al. (2023; 2024; 2025) is a series of studies that LLMs to solve Linear Programming (LP) and Mixed Integer Linear Programming (MILP) problems directly from natural language descriptions. As part of this effort, the authors introduced the NLP4LP dataset, which comprises a diverse collection of LP and MILP problems curated from academic textbooks and lecture materials.

Optibench Yang et al. (2024) is a benchmark for assessing LLMs on LP and MILP modeling, covering diverse problem instances from logistics, scheduling, and resource allocation. It introduces ReSocratic, a data synthesis method that generates high-quality synthetic problems to augment training. Training with such data improves LLMs' ability to interpret natural language, build mathematical models, and produce optimal solutions, making Optibench a state-of-the-art resource for NLP–optimization research.

Beyond this, Bertsimas & Margaritis (2024) applies LLMs to robust and adaptive robust optimization, while the Chain-of-Experts framework Xiao et al. (2023) uses multiple contexts of a single LLM to solve OR tasks via the ComplexOR dataset. Unlike this single-model setup, OR-LLM-Agent Zhang & Luo (2025) offers an end-to-end agentic approach, along with a dataset of 83 real-world OR problems, advancing practical LLM-based optimization.

## 2.2 Reasoning in LLM

Recent advancements in AI reasoning are shifting from System 1 (fast, intuitive thinking) to System 2 (slow, deliberate thinking), as comprehensively reviewed by Li et al. (2025b). A prevalent approach in this context involves leveraging reward models and Monte Carlo Tree Search (MCTS) to backtrack the solution process, using the problem and its answer to guide the exploration, which is then integrated into reinforcement learning (RL). The most closely related works to this paper are those that apply LLMs to solve mathematical problems Shao et al. (2024); Guan et al. (2025); Li et al. (2025a) and coding problems Zhang et al. (2024b); Yu et al. (2024b).

## 2.3 Multi-Agent Collaboration

A substantial body of research has focused on multi-agent collaboration using LLMs. Comprehensive reviews of this area can be found in Sun et al. (2024); Chen et al. (2025). In these studies, multiple LLMs are often integrated to tackle a diverse range of tasks, including reasoning Liu et al. (2024), planning Kannan et al. (2024), coding Hong et al. (2024), financial marketing Gao et al. (2024); Li et al. (2024), education Yu et al. (2024a); Zhang et al. (2024c), and scientific research Baek et al. (2025); Ghafarollahi & Buehler (2024). In many of these works, LLMs are assigned distinct roles to handle different facets of a problem. However, some approaches adopt a framework in which agents perform nearly identical tasks, with the final results being aggregated in the end Cheng et al. (2023). This approach, while effective, should be viewed more as an ensemble method rather than true multi-agent collaboration.

Beyond static role assignment, several studies aim to evolve more dynamic roles and interactions among agents, emphasizing the potential for agents to adapt and specialize over time Hu et al. (2024); Zhuge et al. (2024); Qian et al. (2025). Notably, Qian et al. (2025) proposed a scaling law for LLM-based multi-agent systems, suggesting that such systems may exhibit predictable behavior as they grow in size and complexity. Furthermore, more advanced works explore the use of data generated by multi-agent systems for reinforcement learning, which holds promise for enhancing the capabilities of LLMs in cooperative tasks Park et al. (2025); Nagpal et al. (2025); Zhang et al. (2024a). This direction represents an exciting avenue for future research and development in LLM-based multi-agent systems.

## 3 Methodology

In this section, we present a detailed description of our proposed approach. We begin by outlining the overall pipeline and then highlight two key components: a multi-agent collaboration framework and a multi-armed bandit strategy tailored for the debugging phases. A formal specification of the workflow is provided in Appendix A.

### 3.1 Pipeline

Our agent transforms a natural-language description of an optimization problem into both an executable solver and its corresponding solution. The end-to-end pipeline (illustrated in Figure 1) consists of four sequential stages (**S1**–**S4**). The full set of prompts used in our pipeline is presented in Appendix B.

**S1 Optimization modeling.** This stage translates the natural-language problem statement into a well-defined mathematical optimization problem. The formulation includes identifying the decision variables, specifying the objective function, defining any constraints that must be satisfied, characterizing the problem domain (e.g., continuous, discrete, or mixed), and determining the expected format of the output solution.

**S2 Planning.** In this stage, the agent analyzes the mathematical formulation and proposes multiple candidate strategies. The planner not only selects solvers but also suggests algorithmic strategies and coding insights, with the potential to leverage literature retrieval. In this work, the supported solvers include PuLP, Pyomo, Gekko, OR-Tools, SCIP, MOSEK, IPOPT, and Gurobi, covering a broad spectrum of linear, nonlinear, and mixed-integer optimization techniques.

**S3 Solver code generation.** Given the optimization problem, its mathematical formulation, and a selected solution strategy, this stage generates the corresponding Python solver code. The generated code is required to include data validation and error handling to ensure robustness, solution validation to verify correctness, and informative comments to enhance readability and maintainability.

**S4 Reflective debugging.** The generated code is executed in a runtime environment, where initial execution may often fail due to errors or unexpected behavior. At this stage, the LLM analyzes the code and the resulting error messages, engages in a process of self-reflection to diagnose the underlying issues, and formulates feedback for refining the current strategy. Guided by this reflection, the LLM then iteratively debugs and modifies the code to resolve the identified problems and improve overall reliability.

### 3.2 Multi-Agent Extension

Recent studies have demonstrated that incorporating a multi-agent setting within LLM frameworks can significantly enhance overall performance Hu et al. (2024); Zhuge et al. (2024); Qian et al. (2025). Building on this line of research, we propose to integrate a multi-agent architecture into our system design to enable more effective coordination, specialization, and problem-solving capabilities.

To fully leverage the advantages of a multi-agent system, our pipeline allows different stages to be assigned to different LLMs. Corresponding to the four stages (S1-S4) in our pipeline, we instantiate four roles: **formulator**, **planner**, **coder**, and **code critic**. In addition, we

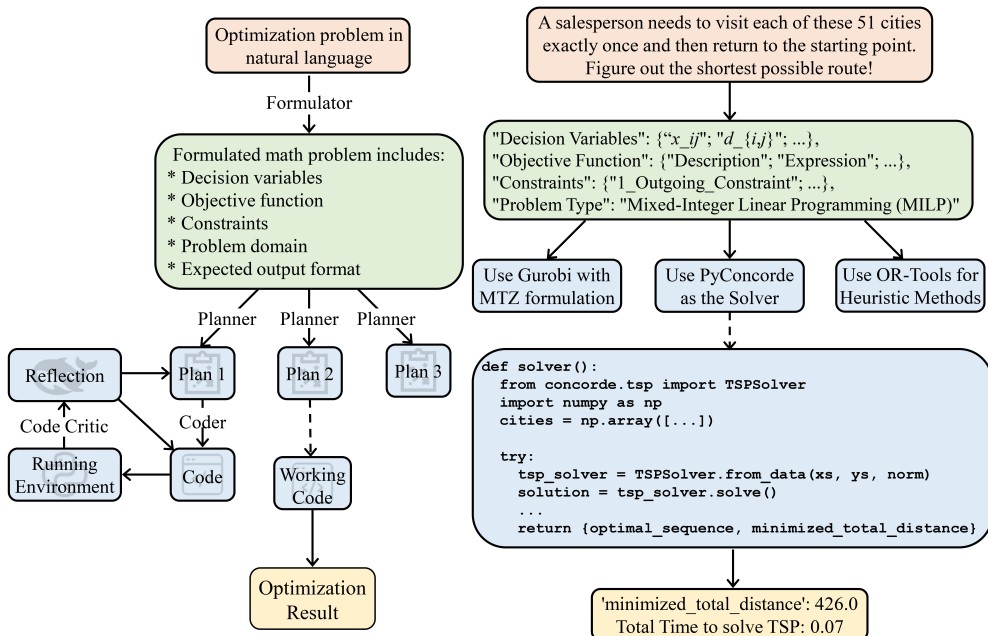

Figure 1: Overview of the OptimAI Pipeline.

introduce two supplementary roles: **decider** and **verifier**. The decider scores plans based on the problem, the proposed plan, and the current code. It helps the system recover from ineffective strategies, as repeated failures prompt the decider to switch to alternative plans. In practice, the probability of selecting the ultimately successful plan on the first attempt is about 39.7%. The verifier ensures that the final outputs satisfy all specified constraints.

Our framework provides flexible configuration options, allowing the entire pipeline to operate with either a single LLM or multiple LLMs assigned to distinct roles. This design not only facilitates the evaluation of individual model performance but also enables us to explore the potential synergistic effects of combining different models for enhanced outcomes.

**Remark 1** *Evidently, reinforcement learning (RL) frameworks leveraging multi-agent architectures tend to exhibit greater robustness and adaptability, particularly in complex or uncertain environments Park et al. (2025). Incorporating such multi-agent RL techniques into our framework could enhance its overall effectiveness, especially in dynamic or error-prone stages such as strategy selection and code debugging. Exploring this integration remains a promising research direction for future research.*

### 3.3 Debug Scheduling as a Multi-Armed Bandit

During the planning phase, the system generates multiple candidate plans to solve the problem. OptimAI then selects the most promising one to initiate implementation, based on evaluations from a separate language model. However, as coding progresses, the initially selected plan may prove ineffective after several rounds of debugging. When this occurs, OptimAI adapts by switching to an alternative plan, mirroring the way humans revise their strategies during problem-solving.

We formulate the problem of selecting which plan to debug next as a multi-armed bandit problem Slivkins et al. (2019). In this abstraction, each plan is treated as an arm. For each plan and its corresponding code, the decider provides a score $\tilde{r}_i$ reflecting how promising it appears and the likelihood of successful debugging. The next arm (plan) is then chosen using the Upper Confidence Bound (UCB) algorithm, where the UCB is defined as

$$\text{UCB}_i := \tilde{r}_i + c\sqrt{\ln(\sum_j n_j)/n_i}, \quad i \in [n],$$

**Algorithm 1** UCB-based Debug Scheduling

**Require:** Problem description
**Ensure:** A working code solution
 1: Generate plans $\{\text{Plan}_1, \ldots, \text{Plan}_n\}$ using planner
 2: Generate code $\text{Code}_i$ for each plan using the coder
 3: Initialize $n_i \leftarrow 1$ for all $i$
 4: **while** no code has succeeded **do**
 5:     Get score $\tilde{r}_i$ from decider for each $(\text{Plan}_i, \text{Code}_i)$
 6:     $\text{UCB}_i \leftarrow \tilde{r}_i + c\sqrt{\ln(\sum_j n_j)/n_i}$
 7:     Select $i^* \leftarrow \arg\max_i(\text{UCB}_i)$
 8:     Debug code $\text{Code}_{i^*}$
 9:     Update $n_{i^*} \leftarrow n_{i^*} + 1$
10: **end while**
11: **return** working code

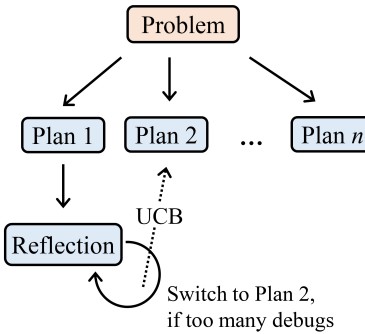

Figure 2: Demonstration of the UCB-based Debug Scheduling.

where $c$ is the exploration coefficient Kocsis & Szepesvári (2006); Auer & Ortner (2010), and $n_i$ is the number of times plan $i$ has been debugged. In the worst case, if the decider assigns the same score to every plan (i.e., lacks discernment), the UCB algorithm naturally reduces to uniform sampling, thereby debugging all plans equally. The full procedure is detailed in Algorithm 1, and Figure 2 illustrates the operation described above.

## 4 EXPERIMENTS

### 4.1 EXPERIMENTAL SETTING

**Datasets** We aim to demonstrate the effectiveness and versatility of our approach by evaluating it on multiple heterogeneous and challenging datasets, as outlined in the following.

    **NLP4LP**. The NLP4LP dataset AhmadiTeshnizi et al. (2024) is a curated collection of 65 LPs aimed at bridging natural language processing and optimization. It includes problem descriptions, parameter data files, and optimal solutions, covering diverse areas like facility location, network flow, scheduling, and portfolio management.

    **Optibench**. The Optibench Yang et al. (2024) contains a diverse set of 605 optimization problems, including linear and nonlinear programming with or without tabular data. This is the first large-scale benchmark to include nonlinear and tabular optimization problems, going beyond the linear programming focus of previous benchmarks.

    **TSPLIB**. The TSPLIB Reinhelt (2014) is a publicly available library of benchmark instances for the Traveling Salesman Problem (TSP), which is known as a standard dataset for evaluating the performance of TSP algorithms.

    **SelfJSP**. The SelfJSP dataset Corsini et al. (2024) is a large-scale benchmark designed for studying neural approaches to solving Job Shop Scheduling Problems (JSP) using supervised learning.

    We also consider some **Set Covering** Problems from IBM ILOG CPLEX Optimization Studio documentation IBM Corporation (2025).

**Baselines** To evaluate the effectiveness of our approach, we compare it against two representative state-of-the-art baselines: OptiMUS AhmadiTeshnizi et al. (2024) and Optibench Yang et al. (2024). We reproduced their code and ran it on GPT-4o to measure the evaluation metrics.

**Evaluation Metrics** We evaluate OptimAI on multiple datasets under the zero-shot prompting setting by measuring the proportion of problems correctly solved in a single call (**Pass@1**). In addition, we assess the quality of the generated solutions using four metrics: Executability, Token Usage, Productivity, and Revisions.

**Executability** is based on human evaluation as defined in Hong et al. (2024), where a score of 4 indicates a completely correct solution; 3 means minor issues; 2 corresponds to barely runnable code with notable problems; and 1 denotes completely non-functional output.

**Token Usage** refers to the average number of tokens consumed by the pipeline to solve a given problem; lower is better.

**Productivity** measures how many lines of code are generated per 1,000 tokens. A higher value indicates better efficiency.

**Revisions** captures the number of debugging attempts to produce executable code.

## 4.2 MAIN RESULT

Table 3: Accuracy comparison between OptimAI and state-of-the-art methods.

| Dataset
Agent | NLP4LP | Optibench Linear | | Optibench Nonlin. | |
| --- | --- | --- | --- | --- | --- |
| | | w/o Tab. | w/ Tab. | w/o Tab. | w/ Tab. |
| OptiMUS | 71.6% | - | - | - | - |
| Optibench | - | 75.4% | 62.5% | 42.1% | 32.0% |
| Ours w/ GPT-4o | 79.1% | 81.2% | 73.8% | 72.0% | 48.0% |
| Ours w/ GPT-4o+o1-mini | **88.1%** | 84.2% | **80.0%** | 77.3% | 56.0% |
| Ours w/ QwQ (by Qwen) | 79.1% | 86.2% | 77.5% | **81.6%** | 50.0% |
| Ours w/ DeepSeek-R1 | 82.1% | **87.4%** | 78.8% | 79.5% | **60.0%** |

All evaluations were conducted under a zero-shot prompting setting. GPT-4o+o1-mini refers to using o1-mini as the planner while employing GPT-4o for all other roles.

Table 3 presents the accuracy of our method on the NLP4LP and Optibench datasets, along with a comparison against previous state-of-the-art approaches. Our approach consistently outperforms prior methods regardless of the underlying LLM employed. Notably, over 99% of the code generated executes without error regardless of the underlying LLM employed. When using DeepSeek-R1, OptimAI achieves an overall accuracy of 82.3% on the Optibench dataset, outperforming the previous best by 8.1 standard deviations.

We further evaluate the executability, token usage, and productivity on the hard subset[1] of the Optibench dataset using GPT-4o. This evaluation is summarized in Table 4. OptimAI outperforms OptiMUS across all metrics and surpasses Optibench in terms of executability. Compared to Optibench, OptimAI incurs a higher token cost but achieves broader problem coverage. Solving a single problem with OptimAI using GPT-4o costs approximately $0.1 on average. The solver usage shows diversity, though Pyomo dominates (48.6%).

Table 4: The Statistical Analysis on OptimAI.

| Statistical Metric | Optibench | OptiMUS | OptimAI |
| --- | --- | --- | --- |
| Executability | 3.4 | 3.1 | **3.5** |
| Token Usage | 955 | 20302 | **18072** |
| Productivity (lines of code/1k tokens) | 45 | 0.72 | **2.32** |

Beyond mathematical programming problems, we further evaluate OptimAI on several representative NP-hard combinatorial optimization problems, including the traveling salesman problem (TSP), job shop scheduling problem (JSP), and set covering problem (SCP). As shown in Table 5, the consistent performance across a diverse set of tasks highlights the robustness and generality of our approach. Notably, it can effectively solve a wide range of challenging combinatorial problems without relying on any problem-specific customization. Detailed examples of how these problems are addressed can be found in Appendix C.

## 4.3 SYNERGISTIC EFFECTS OF COMBINING DISTINCT MODELS

As the Latin proverb goes, *Tres faciunt collegium*[2], combining heterogeneous models can lead to complementary knowledge and improved performance. Our framework supports

---

[1]Hard problems are defined as those requiring more than three debug iterations to reach a runnable state.

[2]Three make company.

Table 5: Generalization of OptimAI across NP-hard combinatorial optimization problems.

| | Math Programming | TSP | JSP | Set Covering |
|---|:---:|:---:|:---:|:---:|
| OptimAI | ✓ | ✓ | ✓ | ✓ |
| OptiMUS | ✓ | ✗ | ✗ | ✗ |
| Optibench | ✓ | ✗ | ✗ | ✗ |

assigning distinct LLMs to different roles, enabling specialization and coordination. To investigate the synergistic effects of mixing heterogeneous LLMs, we assign one model as the planner and another model the remaining roles, and evaluate their performance on the Optibench dataset. The results are presented in Table 6. We observe that combining different LLMs can outperform each individual model. For instance, using Llama 3.3 70B or Gemma 2 27B alone yields accuracies of 59% and 54%, respectively. However, when Gemma 2 27B is assigned as the Planner and Llama 3.3 70B handles the other roles, the accuracy rises to 77%, outperforming any single-model configuration.

Table 6: Synergistic effects of combining heterogeneous LLMs.

| Planner \ Remaining Roles | Llama 3.3 70B | DeepSeek-R1 14B | Gemma 2 27B |
|---|:---:|:---:|:---:|
| Llama 3.3 70B | 59% | 54% | 54% |
| DeepSeek-R1 14B | **68%** | 50% | 41% |
| Gemma 2 27B | **77%** | **59%** | 54% |

## 4.4 Ablation Study

**The Effectiveness of UCB-based Debug Scheduling** To assess the effect of UCB-based debug scheduling, we conduct an ablation study using GPT-4o on the hard subset of the Optibench dataset. As shown in Table 7, disabling the UCB-based debug scheduling results in significantly higher token usage and lower productivity. Specifically, enabling UCB-based debug scheduling reduces token usage by $3.6\times$ and improves productivity by $3.3\times$, while maintaining comparable accuracy and slightly improving executability. These results demonstrate the effectiveness of UCB-based strategies in optimizing debugging efficiency without compromising the quality of outcomes.

Table 7: Ablation study on the impact of UCB-based debug scheduling in OptimAI.

| Evaluation Metric | OptimAI w/o UCB | OptimAI w/ UCB |
|---|:---:|:---:|
| Executability | 3.4 | **3.5** |
| Pass@1 Accuracy | 69% | 69% |
| Token Usage | 64,552 | **18,072** |
| Productivity | 0.70 | **2.32** |

**The Effectiveness of Roles** To understand the impact of different roles and ensure that each component in our pipeline is necessary, we ablated individual stages (roles) and observed the system's performance on the Optibench dataset. As shown in Table 8, removing any single stage leads to a performance drop, confirming the critical contribution of each role to the overall effectiveness of the pipeline. In particular, removing the planner significantly hampers the framework's ability to produce functional code: it requires $4.6\times$ more revisions to reach a runnable state, resulting in a $5.8\times$ drop in productivity. Similarly, omitting the code critic increases the number of revisions by $3.6\times$ and decreases productivity by $3.1\times$. These results highlight the necessity of both high-level planning and post-generation critique in optimizing performance.

**Optimal Exploration Constant** In our prompt, we instruct the decider to return a score between 1 and 10. Assuming a uniform distribution over these scores, the theoretically optimal exploration constant $c$ is $10\sqrt{2}$. However, in practice, the theoretically optimal value may not always yield the best empirical performance. Therefore, we also experimented with

Table 8: Ablation study on roles.

| Formulator | Planner | Code Critic | Revisions | Executability | Productivity |
|:---:|:---:|:---:|:---:|:---:|:---:|
| ✓ | ✓ | ✓ | **1.7** | **3.6** | **6.8** |
| ✗ | ✓ | ✓ | 2.0 | 3.2 | 6.3 |
| ✓ | ✗ | ✓ | 7.8 | 3.1 | 1.2 |
| ✓ | ✓ | ✗ | 6.2 | 3.3 | 2.2 |

$c = 10$ and $c = 20$ on the same set of problems used in Table 7. The results, shown in Table 9, indicate that $10\sqrt{2}$ is a reasonably good choice for the exploration constant in our setting.

Table 9: Comparison for Different Exploration Constants $c$.

| **Exploration Constant $c$** | 10 | $10\sqrt{2}$ | 20 |
|:---|:---:|:---:|:---:|
| Token Usage | 18,672 | **18,072** | 19,989 |
| Accuracy | 69% | 69% | 69% |
| Productivity | 2.29 | **2.32** | 2.1 |

**Optimal Number of Plans**  In our framework, the number of plans generated by the planner, denoted as $n$, is an important hyperparameter. A larger $n$ increases the likelihood of producing at least one feasible plan, which should, in principle, lead to better performance. However, the decider is not particularly strong, so a larger $n$ also increases the chance that the decider selects a non-working plan, potentially degrading performance. We experimented with values of $n$ ranging from 1 to 8 on the same set of problems used in Table 7, and the results are shown in Table 10. We find that $n = 3$ or 4 strikes a good balance. Moreover, we expect that the optimal $n$ will increase as the decider becomes more capable.

Table 10: Comparison for Different Plan Number $n$.

| **Plan Number $n$** | 1 | 2 | 3 | 4 | 5 | 6 | 8 |
|:---|:---:|:---:|:---:|:---:|:---:|:---:|:---:|
| Token Usage | 45,300 | 31,789 | **18,072** | 19,368 | 22,999 | 25,524 | 29,732 |
| Accuracy | 46.2% | 53.8% | 69% | **76.9%** | 69% | 61.5% | 53.8% |

## 5  CONCLUSION

We have presented **OptimAI**, a framework that leverages LLM-powered AI agents to solve optimization problems specified in natural language, achieving superior performance compared to current state-of-the-art methods. Extensive experiments involving four LLMs and five challenging datasets have demonstrated the effectiveness and robustness of our approach. OptimAI's performance is stable across different version of prompts. Looking ahead, we identify several promising directions for future work: (1) reinforcing the framework with RL, especially for fine-tuning the decider component, which has the potential to yield substantial gains with modest computational cost; (2) scaling up OptimAI to tackle large-scale problems that typically require a team of human experts and engineers, moving beyond the current scope, where its performance is comparable to a single skilled programmer. Consequently, OptimAI offers a flexible and extensible foundation for further exploration of multi-agent LLM systems in real-world optimization scenarios.

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

# A FORMAL DESCRIPTION OF METHODOLOGY

We model OptimAI as a sequential decision-making system that manipulates a mutable state memory $\mathcal{S}$, while several specialized roles act upon this memory throughout the workflow. Each role $r \in \mathcal{R}$ is instantiated by a prompt-based agent composed of an underlying language model $F_{\theta_r}$ and a role-specific prompt template $T_r$. The role policy induced by this configuration is written as

$$\pi_r(\mathcal{S}) = F_{\theta_r}(T_r(\mathcal{S})), \tag{1}$$

where $\pi_r$ denotes the behavior of role $r$ when applied to the current state $\mathcal{S}$. We define the role set as

$$\mathcal{R} = \{\text{form, plan, code, critic, dec, ver}\}, \tag{2}$$

corresponding respectively to the formulator, planner, coder, code critic, decider, and verifier. The initial state memory $\mathcal{S}_0$ is a set containing only the problem description $\mathcal{S}_0 = \{\mathcal{P}\}$. Given a current state $\mathcal{S}_t$, OptimAI selects a role $r \in \mathcal{R}$ and applies its policy $\pi_r$ to the state to produce an updated state $\mathcal{S}_{t+1}$. Below, we describe the forward process in detail.

**S1 Optimization Modeling** The formulator constructs a structured representation of the problem based on the initial text description. This process is defined as

$$m_f \leftarrow \pi_{\text{form}}(\mathcal{S}_0), \qquad \mathcal{S}_1 \leftarrow \mathcal{S}_0 \cup \{m_f\}, \tag{3}$$

where $m_f$ is the message generated by the formulator, and $\mathcal{S}_1$ denotes the updated state containing the modeling output.

**S2 Planning** The planner generates multiple high-level strategies (or plans) that describe alternative solution pathways. Each plan corresponds to a different reasoning or programming approach. Formally, we sample $n$ plans from the planner policy

$$\text{plan}_i \sim \pi_{\text{plan}}(\mathcal{S}_1) \quad \text{independently for } i = 1, \ldots, n; \tag{4}$$

$$\mathcal{S}_2^{(i)} \leftarrow \mathcal{S}_1 \cup \{\text{plan}_i\}, \quad i = 1, \ldots, n. \tag{5}$$

Each sampled plan induces a distinct branch in the reasoning trajectory, resulting in $n$ candidate states $\{\mathcal{S}_2^{(1)}, \ldots, \mathcal{S}_2^{(n)}\}$ that can be explored in parallel or selected among in later stages.

**S3 Solver code generation** Each candidate branch $\mathcal{S}_2^{(i)}$ is passed to the coder, which attempts to synthesize an executable program based on the plan and prior context. The code is then executed in a sandbox environment, and both the code and the execution result are appended to the state

$$(\text{code}_i, \text{exec}_i) \leftarrow \pi_{\text{code}}(\mathcal{S}_2^{(i)}), \qquad \mathcal{S}_3^{(i)} \leftarrow \mathcal{S}_2^{(i)} \cup \{\text{code}_i, \text{exec}_i\}. \tag{6}$$

If the execution result indicates success, the updated state is passed to the verifier, which assesses the correctness of the result

$$v_i \leftarrow \pi_{\text{ver}}(\mathcal{S}_3^{(i)}). \tag{7}$$

If $v_i = \texttt{pass}$, the process terminates and returns $\mathcal{S}_3^{(i)}$ as the final output.

**S4 Reflective debugging** If none of the branches pass verification, the system proceeds to debugging. First, each branch is evaluated by the decider, which assigns a quality score based on factors such as code structure, execution logs, or plan plausibility

$$\tilde{r}_{i,t} = \pi_{\text{dec}}(\mathcal{S}_t^{(i)}), \quad i = 1, \ldots, n. \tag{8}$$

To balance exploration and exploitation across candidate plans, we adopt the UCB strategy to select a branch for further refinement. The UCB score for each branch is defined as

$$\text{UCB}_{i,t} = \tilde{r}_{i,t} + c\sqrt{\frac{\ln(\sum_j n_j)}{n_i}}, \tag{9}$$

where $n_i$ denotes the number of times branch $i$ has been previously selected for debugging, and $c > 0$ is an exploration hyperparameter. The branch selected for debugging is

$$i_t^\star = \arg\max_i \text{UCB}_{i,t}. \tag{10}$$

The selected state $\mathcal{S}_t^{(i_t^\star)}$ is first passed to the code critic, which analyzes the prior code and execution and suggests edits or corrections

$$\text{comment}_{i_t^\star} \leftarrow \pi_{\text{critic}}(\mathcal{S}_t^{(i_t^\star)}), \quad \mathcal{S}_{t+1}^{(i_t^\star)} \leftarrow \mathcal{S}_t^{(i_t^\star)} \cup \{\text{comment}_{i_t^\star}\}. \tag{11}$$

The updated state $\mathcal{S}_{t+1}^{(i_t^\star)}$ is then reapplied to the coder for regeneration and re-execution

$$(\text{code}_{i_t^\star}, \text{exec}_{i_t^\star}) \leftarrow \pi_{\text{code}}(\mathcal{S}_{t+1}^{(i_t^\star)}), \quad \mathcal{S}_{t+1}^{(i_t^\star)} \curvearrowleft \{\text{code}_{i_t^\star}, \text{exec}_{i_t^\star}\}. \tag{12}$$

Here, the symbol $\curvearrowleft$ denotes replacement of prior entries for code and execution in the state, as opposed to appending new messages. Finally, we update the selection count

$$n_{i_t^\star} \leftarrow n_{i_t^\star} + 1. \tag{13}$$

Finally, the updated state $\mathcal{S}_{t+1}^{(i_t^\star)}$ is passed to the verifier. If verification succeeds, the system terminates and returns $\mathcal{S}_{t+1}^{(i_t^\star)}$ as the final output. Otherwise, the system repeats the debugging process beginning from Eq. equation 8, continuing until verification succeeds or a predefined iteration limit $T_{\max}$ is reached.

## B  PROMPT EXAMPLES

### Formulator

```
You are an expert in Optimization Modeling.
Analyze the following optimization problem and extract its key components:

### Optimization Problem:
{state["messages"][0].content}

### Task:
First, analyze the optimization problem and then provide the following components:
- Decision Variables: Include their types and domains. Make their names descriptive
- Objective Function: Specify the objective expression.
- Constraints: List all constraints.
- Problem Type: Specify the type of optimization problem (e.g., Linear Programming, Mixed-Integer \
        Linear Programming, Non-Linear Programming, Mixed-Integer Non-Linear Programming, Quadratic \
        Programming, etc.). This information needs to be very precise and accurate.
- Table Description: Provide an easy-to-understand description of the table if there is a table \
        present in the question, else return an empty string. Note that the table description should \
        effectively describe each and every value inside the table.

### Response Format:
Return your response as a JSON object with the keys:
"Decision Variables", "Objective Function", "Constraints", "Problem Type", and "Table Description".
"""
    else:
          # Use the feedback prompt if human feedback is available
          prompt = f"""You are an expert in Optimization Modeling.
Revise the extracted components of the following optimization problem based on human feedback.

### Optimization Problem:
{state["messages"][0].content}

### Current Components:
- Decision Variables: {state["components"]["decision_variables"]}
- Objective Function: {state["components"]["objective_function"]}
- Constraints: {state["components"]["constraints"]}
- Problem Type: {state["components"]["problem_type"]}
- Table Description: {state["componenets"]["table_description"]}

### Human Feedback:
{state["components"]["user_feedback"]}

### Task:
Revise the components based on the feedback while ensuring accuracy.

### Response Format:
Return your response as a JSON object enclosed in ```json``` tags with the keys:
"Decision Variables", "Objective Function", "Constraints", "Problem Type", and "Table Description".
```

### Planner

```
You are an Optimization solver code planning expert.
Your responsibility is to provide three best strategies to implement the source code to solve an \
        Optimization Problem.

First, carefully analyze the optimization problem, try to understand the type of problem it is, \
        and, if given, examine the user recommendations for specific requirements, preferences, or \
        domain knowledge that should influence your solution approach.
Next, thoroughly analyse the optimization model components, constraints, variables, and objective \
        function.
Then, understand the available solvers and modeling tools we have = {Available_Tools}.
Analyse the strengths and weaknesses of each solver and decide the ones that align best with the \
        given problem and its type.

Then think about the three most effective strategies that can be implemented to generate the solver \
        code for this problem.

Each strategy must include the following:
1) suitable Optimization solver for the task (You can only use the solvers and the modelling tools \
        from the list we provided you)
2) details about the algorithm to implement
3) any other information to be kept in mind while implementing the solver code. (Do not include \
        information on solver installation)
Make sure your strategies are elaborated in depth, which will enable better code generation using \
        them.

Only make use of reliable sources, such as academic papers and official documentations, during your \
        tool calls to the tavily_tool to arrive at your conclusion
However, do not include the sources in your final strategies.
```

```
Only after completing this thorough analysis, provide your response.

### Optimization Problem:
{state["messages"][0].content}

### User recommendations (if any)
{UserFeedbackRecord.user_recommendations}

### Optimization Model
{state["components"]}

### Response Format:
Your response should ONLY contain a list of the three strategies without any additional formatting.
```

## Decider

```
You are an Optimization solver code implementation expert.
Your responsibility is to provide the scores (from 1 to 10, with 10 being the best) for the \
        strategies to implement a solver code for an optimization problem.

First, carefully analyse the optimization problem and its components.
Then, understand and analyse the provided strategies to implement the code to solve the \
        optimization problem.
Finally, score these strategies and provide your response in the requested format.

### Optimization Problem:
{state["messages"][0].content}

### Strategies to implement the solver code
{state["messages"][-1].content}

### Task:
Rate the strategies to implement the code to solve the given optimization problem from 1 to 10, \
        with 10 being the best:
- Consider which strategy would lead to the most efficient code that would give us the most \
        accurate result.
- The ratings should be relative to each other

Do not modify any strategy given to you.

### Response Format:
Return your response as a JSON object with the following keys:
"Strategy1", "Strategy2", and "Strategy3".
"Strategy1" should map to the score for the provided Strategy1, "Strategy2" should map to the score \
        for the provided Strategy2, and "Strategy3" should map to the score for the provided Strategy3.
```

## Coder

```
You are a Python coding expert for solving Optimization problems.
Your responsibility is to provide the Python code to get the solution to a given optimization \
        problem.

First, carefully analyze the optimization problem and the optimization model components, \
        constraints, variables, and objective function.
Then, understand and analyse the given strategy to generate the code to solve that problem and the \
        requirements specified for the desired code.

Then think about the most effective way to generate the solver code using the provided strategy for \
        this problem.

Only after completing this thorough analysis, provide the desired Python code.

 ### Optimization Problem:
{state["messages"][0].content}

### Optimization Model
{state["components"]}

### Strategy to Implement the Solver Code
{state[active_branch][0]["strategy"]}

### Requirements For the Python Code
The code must only contain a single function named 'solver'.
Generate code that:
    - Follows the given strategy.
    - Includes all necessary imports
    - Implements proper data validation and error handling
    - Creates all variables with correct types and bounds
    - Defines the objective function exactly as shown
    - Implements all constraints from the model
```

```
        - Solves the model and checks the solution status
        - Format output as a dictionary with variable values
        - Report errors clearly by returning the "error" in the dictionary
        - Returns and prints both the optimal solution and the objective value
        - Includes the arguments in the function definition, if any
        - Handles unit conversions appropriately
        - Validates that output
        - Includes comments

Make sure you are returning the value requested in the question.
Focus on the accuracy of the final solution and make sure it satisfies all the requirements given \
        in the optimization problem.

Make sure that the final answer logically makes sense, i.e., a variable in the solution does not \
        have a decimal value when it logically can not.

The code you return, when run by itself using Python's exec() function, should give the final \
        solution to the optimization problem.
Basically, your code should be self-sufficient to run by itself and give the final solution to the \
        given optimization problem.
There should NOT be any need to make any adjustments to your code, like calling a function, putting \
        parameter values, etc., to get the desired solution to the optimization problem.

### Response Format:
Your response should ONLY contain the Python code, which only contains a single function named \
        'solver', with no additional formatting.
```

## Code Critic

```
You are an expert at analyzing Python optimization solver code.
Your job is to provide feedback to debug the given code to solve an optimization problem.
Also, you have to score the code on a scale of 1 to 10 on how likely it is to get debugged by LLM \
        API calls.
First, carefully analyze the provided Optimization Problem, Optimization Model, and the Strategy to \
        Implement the Solver Code. Then, analyse the provided Code to solve this problem and the \
        error returned to understand the problem.
Finally, understand your task and provide your response in the requested format. Make sure that the \
        constraints in the question and the model are followed precisely; this is very important.

### Optimization Problem:
{state["messages"][0].content}

### Optimization Model
{state["components"]}

### Strategy to Implement the Solver Code
{state[active_branch][len(state[active_branch]) - 1]["strategy"]}

### Code to Solve the Problem
{solver_code}

### Error Returned by the Code
{error_msg}

### Task
You need to provide feedback to help debug the code to generate the requested solution for a given \
        optimization problem.

Keep in mind that the feedback you provide should be such that the debugged code should be able to \
        execute by itself using Python's exec() function without any additional steps.
Only provide the feedback to help debug the code in simple English; do not provide any debug code \
        in your feedback.

Lastly, also provide the score, between 1 to 10, on the likelihood of the given code being \
        successfully debugged using successive LLM API Calls. (Note 1 being the lowest and 10 being \
        the highest)

### Response Format

Your response must be a JSON object with the keys: "feedback" and "score".
The key "feedback" should map to the valid feedback to help debug the code. It should contain only \
        the feedback to help debug the code without any formatting.
The "score" key should map to the score of the likelihood of this code being debugged by successive \
        api calls.
```

## Code Debug

```
You are a Python expert at debugging code for solving Optimization problems.
Your responsibility is to debug the provided Python code to get the solution for a given \
        optimization problem.
```

```
First, carefully analyze the optimization problem and the optimization model components, \
        constraints, variables, and objective function.
Then, understand and analyse the given code, the strategy to generate the code to solve that \
        problem, and the requirements specified for the desired code.
Then analyse the error and the feedback to debug the code.

Then, think about debugging the given code while maintaining the strategy being employed and the \
        requirements for the code.

Only after completing this thorough analysis, provide the desired debugged Python code.
The code must only contain a single function named 'solver'.

### Optimization Problem:
{state["messages"][0].content}

### Optimization Model
{state["components"]}

### Strategy to Implement the Solver Code
{state[active_branch][current_length - 1]["strategy"]}

### Solver Code
{state[active_branch][current_length - 1]["code"]}

### Error
{state[active_branch][current_length - 1]["error"]}

### Feedback to debug the code
{state[active_branch][current_length - 1]["critique"]}

### Requirements for the debugged Python Code
Generate debugged code without altering the characteristics of the provided code.
The code must only contain a single function named 'solver'.
Essentially, the debugged code should:
    - Follows the given strategy.
    - Includes all necessary imports
    - Implements proper data validation and error handling
    - Creates all variables with correct types and bounds
    - Defines the objective function exactly as shown
    - Implements all constraints from the model
    - Solves the model and checks the solution status
    - Format output as a dictionary with variable values
    - Report errors clearly by returning the "error" in the dictionary
    - Returns and prints both the optimal solution and the objective value
    - Includes the arguments in the function definition, if any
    - Handles unit conversions appropriately
    - Validates that output
    - Includes comments

Focus on the provided feedback to help debug the code.
Make sure you are returning the value requested in the question.
Focus on the accuracy of the final solution and make sure it satisfies all the requirements given \
        in the optimization problem.

Make sure that the final answer logically makes sense, i.e., a variable in the solution does not \
        have a decimal value when it logically can not.

The code you return, when run by itself using Python's exec() function, should give the final \
        solution to the optimization problem.
Basically, your code should be self-sufficient to run by itself and give the final solution to the \
        given optimization problem.
There should NOT be any need to make any adjustments to your code, like calling a function, putting \
        parameter values, etc., to get the desired solution to the optimization problem.

### Response Format:
Your response should ONLY contain the Python code, which only contains a single function named \
        'solver', with no additional formatting.
```

## Verifier

```
You are an expert at analyzing whether a certain optimization code evaluates the optimization \
        problem successfully by fulfilling all the constraints.
Your job is to provide Python code that verifies whether a given optimization problem fulfills a \
        set of constraints.

First, carefully analyze the provided Optimization Problem, Optimization Model, the Strategy to \
        Implement the Solver Code, and the solver code itself.
Then, analyse the final result produced by the code.

Finally, understand your task and provide a Python code to verify if the results follow the problem \
        constraints to solve the problem.

### Optimization Problem:
```

```
{state["messages"][0].content}

### Optimization Model
{state["components"]}

### Strategy to Implement the Solver Code
{state[active_branch][len(state[active_branch]) - 1]["strategy"]}

### Code to Solve the Problem
{state['final_code']}

### Result produced by the code
{state['final_output']}

### Task
You need to provide Python code to check if the given solution produced by the given code adheres \
        to the constraints in the provided optimization problem.
The code must ONLY contain a single function named 'solver'.
Generate code that:
- Includes all necessary imports, inside the function
- Implements proper data validation and error handling
- Creates all variables with correct types and bounds
- Implements all constraints from the model and the problem
- Format output as a dictionary with the following specifications:

The dictionary should have just a single key named "evaluation".
It should map to "correct" if the solution follows the set of constraints.
Else, it should map to a comma-separated list of variable names that do not follow the set of \
        constraints.

The code you return, when run by itself using Python's exec() function, should give the final \
        solution to the optimization problem.
Basically, your code should be self-sufficient to run by itself and give the final solution to the \
        given optimization problem.
There should NOT be any need to make any adjustments to your code, like calling a function, putting \
        parameter values, etc., to get the desired solution to the optimization problem.

### Response Format
Your response should ONLY contain the Python code, which only contains a single function named \
        'solver', with no additional formatting.
```

# C  CASE STUDY

## C.1  TRAVELING SALESPERSON PROBLEMS WITH 280 CITIES

This is the a280 problem from the popular TSPLIB dataset Reinhelt (2014). The total time to solve TSP is 0.82 seconds.

---

**Optimization Modeling**

```
{
  "Decision Variables": {
    "x_ij": {
      "Description": "Binary variable indicating whether the edge from node i to node j is included \
          in the tour.",
      "Type": "Binary",
      "Domain": "x_ij \\in \\{0, 1\\} for all i, j \\in \\{1, 2, ..., 280\\}"
    },
    "u_i": {
      "Description": "Continuous variable used for subtour elimination (Miller-Tucker-Zemlin \
          formulation) representing the position of node i in the tour.",
      "Type": "Continuous",
      "Domain": "2 \\leq u_i \\leq 280 for all i \\in \\{2, 3, ..., 280\\}"
    }
  },
  "Objective Function": {
    "Description": "Minimize the total Euclidean distance of the tour.",
    "Expression": "Minimize \\sum_{1 \\leq i \\neq j \\leq 280} (d_ij * x_ij)",
    "Where": "d_ij is the Euclidean distance between node i and node j."
  },
  "Constraints": {
    "1_Flow_Conservation": {
      "Description": "Ensure that exactly one outgoing edge leaves each node.",
      "Expression": "\\sum_{1 \\leq j \\leq 280, j \\neq i} x_ij = 1 for all i \\in \\{1, 2, ..., \
          280\\}"
    },
    "2_Flow_Conservation": {
      "Description": "Ensure that exactly one incoming edge enters each node.",
      "Expression": "\\sum_{1 \\leq i \\leq 280, i \\neq j} x_ij = 1 for all j \\in \\{1, 2, ..., \
          280\\}"
    },
    "3_Subtour_Elimination": {
      "Description": "Eliminate subtours using the Miller-Tucker-Zemlin (MTZ) constraints.",
      "Expression": "u_i - u_j + 280 * x_ij \\leq 279 for all i, j \\in \\{2, 3, ..., 280\\}, i \
          \\neq j"
    },
    "4_Start_Node": {
      "Description": "Define the starting node of the tour.",
      "Expression": "u_start = 1",
      "Note": "Assuming a specific starting node is selected, otherwise this constraint can be \
          omitted or adjusted."
    }
  },
  "Problem Type": "Mixed-Integer Linear Programming (MILP)"
}
```

---

**Plans**

```
1)      Asks the user if they want to input suggestions for the strategies to implement the solver \
        code. Human inputs "I want the output to be the most optimal solution to the given TSP, and I \
        also want the code to be feasible to run on my laptop, a Macbook Air".
2)      Planner Agent makes the following tool calls to search the web:
query: Optimal code for Symmetric TSP problems
query: pyconcorde solving TSP Concorde documentation
query: Using OR-Tools for TSP
3)      Finally, the Planner Agent provides the following strategies:

1) Utilize the Pyconcorde Solver:
   - Suitable Solver: Pyconcorde
   - Algorithm Details: Pyconcorde is a Python wrapper for the Concorde TSP Solver. Concorde is a \
        highly optimized and specialized solver for the TSP based on branch-and-cut algorithms. \
        It's particularly suitable for this problem due to its efficiency in handling large \
        instances of TSP by dynamically adding constraint cuts to eliminate fractional solutions \
        and subtours within the branch-and-bound tree.
   - Additional Information: Ensure input data is correctly formatted for Pyconcorde. It requires \
        the adjacency matrix or list of city coordinates. Consider pre-processing the distances \
        using Euclidean calculation to speed up data provision. Pyconcorde will handle solving \
        with minimal setup complexity, suitable for a 280-node problem.

2) Implement with OR-Tools and SCIP:
   - Suitable Solver: OR-Tools with SCIP backend
```

```
        - Algorithm Details: OR-Tools supports TSP through routing solvers utilizing the Constraint \
            Programming and MILP approaches. SCIP, accessed via OR-Tools, employs effective numerics \
            and heuristics along with branch-and-bound strategies. With constraint propagation and \
            optional LNS solvers in OR-Tools, this strategy converges quickly on optimum solutions.
        - Additional Information: Leverage OR-Tools to set up TSP in a routing problem context. Use \
            distance callbacks to integrate the Euclidean distance matrix. The use of parallel \
            processing and heuristic guidance within SCIP via OR-Tools enhances performance for \
            optimization, potentially complemented by metadata to improve node ordering and path \
            searches.

3) Use Gurobi for MILP Modeling:
    - Suitable Solver: Gurobi
    - Algorithm Details: Gurobi is a powerful MILP solver with robust algorithms for solving \
        large-scale MIPs. Use the Miller-Tucker-Zemlin (MTZ) sub-tour elimination constraints. \
        Gurobi's advanced presolve techniques, cutting planes, and node-level parallel executions \
        make it apt for exact TSP modeling as per MILP formulations.
    - Additional Information: Set up the problem with binary decision variables for routing between \
        cities and continuous variables for subtour elimination based on the MTZ model. Ensure \
        model environment settings in Gurobi exploit multi-core processing and memory efficiently. \
        Preprocessing steps should involve constructing accurate distance matrices and tuning \
        solver parameters like MIPGap, NodeLimit, and heuristics to balance speed and solution \
        precision.
```

### Decider

```
Decider Agent:

The decider agent ranks the strategies in the following order:

1)Utilize the Pyconcorde Solver
2)Implement with OR-Tools and SCIP
3)Use Gurobi for MILP Modeling
```

### Final Result

```
{'optimal_route': [1, 2, 242, 243, 244, 241, 240, 239, 238, 237, 236, 235, 234, 233, 232, 231, 246, \
    245, 247, 250, 251, 230, 229, 228, 227, 226, 225, 224, 223, 222, 221, 220, 219, 218, 217, \
    216, 215, 214, 213, 212, 211, 210, 209, 208, 207, 206, 205, 204, 203, 202, 201, 200, 144, \
    145, 199, 198, 197, 196, 195, 194, 193, 192, 191, 190, 189, 188, 187, 186, 185, 184, 183, \
    182, 181, 176, 180, 179, 150, 178, 177, 151, 152, 156, 153, 155, 154, 129, 128, 127, 126, \
    125, 124, 123, 122, 121, 120, 119, 157, 158, 159, 160, 175, 161, 162, 163, 164, 165, 166, \
    167, 168, 169, 170, 171, 172, 173, 174, 107, 106, 105, 104, 103, 102, 101, 100, 99, 98, 97, \
    96, 95, 94, 93, 92, 91, 90, 89, 109, 108, 110, 111, 114, 113, 112, 88, 87, 84, 83, 82, 81, \
    80, 79, 78, 77, 75, 76, 74, 73, 72, 71, 70, 67, 69, 68, 58, 57, 56, 55, 54, 53, 52, 51, 50, \
    49, 48, 47, 46, 45, 44, 59, 63, 64, 66, 65, 85, 86, 116, 115, 117, 118, 62, 61, 60, 43, 42, \
    41, 40, 39, 38, 37, 36, 35, 34, 33, 32, 31, 30, 29, 28, 27, 26, 22, 25, 23, 24, 14, 15, 13, \
    12, 11, 10, 8, 9, 7, 6, 5, 4, 277, 276, 275, 274, 273, 272, 271, 16, 17, 18, 19, 20, 21, 130, \
    131, 132, 133, 134, 270, 269, 135, 136, 268, 267, 137, 138, 139, 149, 148, 147, 146, 143, \
    142, 141, 140, 266, 265, 264, 263, 262, 261, 260, 259, 258, 257, 254, 253, 252, 255, 256, \
    249, 248, 278, 279, 3, 280], 'total_distance': 2579.0}
```

## C.2 Job Scheduling Problem with 20 Jobs and 5 Machines

This is the la11.jsp problem from SelfJSP Corsini et al. (2024).

### Optimization Modeling

```
{
  "Decision Variables": {
    "Start_Time_jk": {
      "Description": "Start time of the k-th operation of job j",
      "Type": "Continuous",
      "Domain": "Non-negative real numbers"
    },
    "Makespan": {
      "Description": "Overall completion time of all jobs",
      "Type": "Continuous",
      "Domain": "Non-negative real numbers"
    }
  },
  "Objective Function": {
    "Expression": "Minimize Makespan"
```

```
    },
  "Constraints": [
    {
      "Description": "Precedence constraints within each job",
      "Mathematical_Form": "For each job j and for each operation k > 1, Start_Time_jk \\geq \
            Start_Time_j(k-1) + Processing_Time_j(k-1)"
    },
    {
      "Description": "No overlap on each machine",
      "Mathematical_Form": "For any two operations (j1, k1) and (j2, k2) assigned to the same \
            machine m, either Start_Time_j1k1 + Processing_Time_j1k1 \\leq Start_Time_j2k2 or \
            Start_Time_j2k2 + Processing_Time_j2k2 \\leq Start_Time_j1k1"
    },
    {
      "Description": "Completion time of each job does not exceed Makespan",
      "Mathematical_Form": "For each job j, Start_Time_jK + Processing_Time_jK \\leq Makespan"
    },
    {
      "Description": "Start times are non-negative",
      "Mathematical_Form": "For all jobs j and operations k, Start_Time_jk \\geq 0"
    }
  ],
  "Problem Type": "Mixed-Integer Linear Programming (MILP)"
}
```

## Plans

```
1) Strategy 1:
   - Suitable Optimization Solver: CP-SAT Solver from OR-Tools
   - Algorithm: The CP-SAT solver utilizes a combination of SAT (satisfiability testing) and \
         constraint programming approaches to efficiently solve the scheduling problem. Model the \
         problem using interval variables for each operation to represent their start times and \
         processing durations. Leverage no-overlap constraints within the solver to handle the \
         machine assignment to avoid conflicts, and enforce job precedence constraints to maintain \
         operation order within jobs.
   - Other Considerations: Keep in mind the solver's ability to handle large search spaces with \
         advanced search heuristics, and utilize the solver's objective function capabilities to \
         minimize the makespan effectively, while using decision variable strategies like \
         warm-start or hinting when appropriate to speed up the solution process.

2) Strategy 2:
   - Suitable Optimization Solver: OR-Tools (Using its Job Shop Scheduling and Integer Linear \
         Programming functionalities)
   - Algorithm: Embed the operations and jobs within the Job Shop Scheduling framework, which is \
         specifically designed for handling machine scheduling problems with constraints. Use \
         integer programming to model the decision variables as integer values leading to efficient \
         computation of start times and precedence. Explore OR-Tools' symmetry breaking constraints \
         and specialized cutting planes for optimizing the schedule.
   - Other Considerations: Make full use of OR-Tools' scalable performance on large instances of \
         scheduling problems. Consider breaking down the problem into smaller subproblems where \
         possible and apply iterative solving techniques like Lagrangian relaxation to ensure more \
         consistent convergence to optimal solutions.

3) Strategy 3:
   - Suitable Optimization Solver: Pyomo with Gurobi
   - Algorithm: Use Pyomo's flexible modeling environment to express the MILP model of the \
         scheduling problem, including variables, objectives, and constraints as defined. Integrate \
         with Gurobi's solver capabilities, which include powerful branch-and-cut algorithms to \
         optimize the makespan. This combination allows fine-tuning of variable bounds and solver \
         parameters to efficiently explore the solution space.
   - Other Considerations: Gurobi's optimization suite has extensive parameter tuning \
         possibilities, like tweaking MIP gap tolerances, setting relative and absolute termination \
         criteria, and using Gurobi's heuristics to achieve faster solutions. While Pyomo allows \
         for simplicity in expressing complex relations, ensure computational resource constraints \
         are accounted for when dealing with the CPU- and time-intense parts of the solving \
         process, especially for evaluating inherent delays in large-scale problems.
```

## Decider

```
The decider agent ranks the strategies in the following order:

1)CP-SAT Solver from OR-Tools
2)OR-Tools (Using its Job Shop Scheduling and Integer Linear Programming functionalities)
3)Suitable Optimization Solver: Pyomo with Gurobi
```

## Code Critic

```
Ensure that the `cp_model` is correctly imported by verifying that the line `from \
    ortools.sat.python import cp_model` is present and properly placed before any usage of \
    `cp_model`. Make sure there are no typos in the import statement and that it is accessible \
    within the scope where `cp_model` is used.
```

## Final Result

```
{'Makespan': 1222.0, 'Start_0_0': 66, 'Start_0_1': 298, 'Start_0_2': 627, 'Start_0_3': 689, \
    'Start_0_4': 917, 'Start_1_0': 680, 'Start_1_1': 744, 'Start_1_2': 842, 'Start_1_3': 1040, \
    'Start_1_4': 1138, 'Start_2_0': 436, 'Start_2_1': 458, 'Start_2_2': 509, 'Start_2_3': 674, \
    'Start_2_4': 1015, 'Start_3_0': 0, 'Start_3_1': 66, 'Start_3_2': 438, 'Start_3_3': 798, \
    'Start_3_4': 932, 'Start_4_0': 353, 'Start_4_1': 637, 'Start_4_2': 822, 'Start_4_3': 913, \
    'Start_4_4': 1039, 'Start_5_0': 325, 'Start_5_1': 466, 'Start_5_2': 535, 'Start_5_3': 627, \
    'Start_5_4': 788, 'Start_6_0': 0, 'Start_6_1': 165, 'Start_6_2': 242, 'Start_6_3': 371, \
    'Start_6_4': 458, 'Start_7_0': 242, 'Start_7_1': 325, 'Start_7_2': 666, 'Start_7_3': 707, \
    'Start_7_4': 1162, 'Start_8_0': 404, 'Start_8_1': 500, 'Start_8_2': 701, 'Start_8_3': 745, \
    'Start_8_4': 1166, 'Start_9_0': 93, 'Start_9_1': 223, 'Start_9_2': 423, 'Start_9_3': 560, \
    'Start_9_4': 936, 'Start_10_0': 258, 'Start_10_1': 353, 'Start_10_2': 781, 'Start_10_3': \
    1012, 'Start_10_4': 1103, 'Start_11_0': 517, 'Start_11_1': 540, 'Start_11_2': 931, \
    'Start_11_3': 1054, 'Start_11_4': 1081, 'Start_12_0': 0, 'Start_12_1': 172, 'Start_12_2': \
    858, 'Start_12_3': 1116, 'Start_12_4': 1183, 'Start_13_0': 145, 'Start_13_1': 319, \
    'Start_13_2': 772, 'Start_13_3': 1088, 'Start_13_4': 1116, 'Start_14_0': 429, 'Start_14_1': \
    448, 'Start_14_2': 535, 'Start_14_3': 635, 'Start_14_4': 1009, 'Start_15_0': 91, \
    'Start_15_1': 778, 'Start_15_2': 815, 'Start_15_3': 922, 'Start_15_4': 968, 'Start_16_0': \
    527, 'Start_16_1': 638, 'Start_16_2': 796, 'Start_16_3': 853, 'Start_16_4': 931, \
    'Start_17_0': 549, 'Start_17_1': 745, 'Start_17_2': 843, 'Start_17_3': 856, 'Start_17_4': \
    1056, 'Start_18_0': 81, 'Start_18_1': 189, 'Start_18_2': 329, 'Start_18_3': 707, \
    'Start_18_4': 1054, 'Start_19_0': 0, 'Start_19_1': 100, 'Start_19_2': 145, 'Start_19_3': 223, \
    'Start_19_4': 992}
```

