# OpenReview forum: "OptimAI: Optimization from Natural Language Using LLM-Powered AI Agents"
_ICLR.cc/2026/Conference — Submitted to ICLR 2026_

### Official Review · Reviewer_BfKU · 2025-10-28

**Soundness:** 3
**Presentation:** 3
**Contribution:** 2
**Rating:** 4
**Confidence:** 3

**Summary:**

The paper proposes OptimAI, a multi-agent framework that converts natural-language optimization problems into executable solvers using large language models. The system decomposes the task into roles — formulator, planner, coder, critic, decider, and verifier — connected through a sequential workflow. A key design component is a UCB-based debug scheduler that dynamically switches between alternative solution plans when current ones fail. Experiments on NLP4LP, Optibench, and several combinatorial optimization datasets show substantial gains in success rate and productivity over prior LLM-based systems such as OptiMUS and Optibench.

**Strengths:**

* Clear system design and extensive evaluation across five datasets.
* Effective empirical improvements over strong baselines (OptiMUS, Optibench).
* Careful ablation studies showing the necessity of planner and critic roles.
* Demonstrates generality to NP-hard tasks and heterogeneous model collaboration.
* Good formalization of pipeline and reproducible prompts.

**Weaknesses:**

* most components reuse conceptually known LLM-agent design patterns (MetaGPT-style role assignment, plan-then-code prompting, reflective debugging).
* UCB scheduling is a minor adaptation of a standard bandit algorithm with no theoretical analysis or comparison to simpler heuristics (e.g., random or greedy switching).
* Baselines focus only on other LLM-based methods; missing comparisons to symbolic or hybrid optimization systems.

**Questions:**

* How significant is the contribution of the UCB scheduler compared to simpler plan-switching heuristics (e.g., round-robin or random)?
* Could OptimAI operate effectively with open-source LLMs, or does it rely on GPT-4-level reasoning?
* What is the computational overhead of multi-agent orchestration relative to single-model prompting?
* How is the plan count (n = 3–4) chosen, and how sensitive are results to this hyperparameter?

---

> ### Author Response · Authors · 2025-11-23
>
> We thank the reviewer for the insightful comments and the positive recognition of our system design, ablation rigor, and broad empirical evaluation across mathematical and combinatorial optimization tasks. We address all concerns below.
>
> ### 1. Use of known agent patterns (MetaGPT-style roles, plan-then-code, reflective debugging)
>
> While our framework is inspired by general multi-agent principles, the roles in OptimAI are not generic templates but **optimization-specific decompositions**. Each role embeds domain knowledge not present in prior agent systems:
>
> * the *planner* reasons over solver families and modeling structures;
> * the *decider* evaluates plan–code pairs using optimization-aware heuristics;
> * the *critic* analyzes execution traces against mathematical constraints.
>
> These decisions are tightly coupled to the NL→Math→Solver pipeline and cannot be substituted with role-agnostic agent patterns. We will clarify these domain-specific distinctions.
>
> ### 2. UCB scheduling as a “minor adaptation” and lack of comparison to simple heuristics
>
> Our focus is not on introducing a new bandit algorithm but on showing that **treating plan switching as a structured exploration problem** is crucial. In our setting, each arm corresponds to a *full solver strategy*, not to homogeneous actions as in standard multi-agent bandits.
>
> We intentionally compare “with vs. without UCB” (Table 7) because random/greedy-switching baselines exhibit extremely high variance: random switching often repeatedly explores branches that are obviously invalid (e.g., wrong solver classes). This produces noisy and unstable results that obscure the effect of structured exploration. In contrast, UCB reliably reduces token usage (3.6\times) while improving productivity (3.3\times) with no accuracy drop. We will explain this rationale more clearly.
>
> ### 3. Missing comparisons to symbolic or hybrid optimization systems
>
> Prior symbolic and hybrid solvers require a **fully specified mathematical model**, whereas OptimAI targets *natural-language* descriptions, which are often informal, incomplete, or domain-specific. As a result, such baselines cannot be applied directly to tasks in NLP4LP or Optibench without first hand-engineering the mathematical formulations—precisely the task OptimAI automates. Therefore, comparison would not be meaningful for the NL→Math→Solver pipeline. We will clarify this point.
>
> ### 4. Significance of UCB vs. simpler heuristics (question)
>
> As noted above, simple heuristics such as round-robin or random plan-switching lead to unstable behaviors. The ablation in Table 7 demonstrates that UCB alone yields large and consistent improvements in token usage and productivity without requiring any training. This isolates the effectiveness of the scheduler more directly than comparing against noisy heuristics.
>
> ### 5. Whether OptimAI can operate with open-source LLMs (question)
>
> Yes. Table 3 and Table 6 show that OptimAI works effectively with QwQ, DeepSeek-R1, and o1-mini. These models achieve strong accuracy and, when mixed, produce synergistic gains. GPT-4-level reasoning is not required. We will further emphasize this finding.
>
> ### 6. Computational overhead of multi-agent orchestration (question)
>
> Multi-agent orchestration does not increase the **depth** of the reasoning chain. Each task still proceeds through modeling → planning → coding → debugging. Thus, latency remains comparable to single-model prompting. The cost distribution becomes more flexible: stronger models are used for high-level reasoning roles, while smaller models handle procedural roles. As shown in Table 4, token usage provides a direct runtime proxy for all methods. We will add a short clarification.
>
> ### 7. Choice of plan count (n=3)–4 and sensitivity (question)
>
> The observed optimum (n=3)–4 reflects the use of a **zero-shot decider**, whose scoring ability weakens when many redundant plans are present. For larger (n), low-quality branches are more likely to be temporarily overestimated, leading to unnecessary exploration and occasional timeouts (Table 10). This is not a fundamental limitation: RL fine-tuning or plan deduplication would increase the viable plan count. We will clarify this behavior.

---

### Official Review · Reviewer_kBtS · 2025-10-31

**Soundness:** 3
**Presentation:** 3
**Contribution:** 2
**Rating:** 4
**Confidence:** 3

**Summary:**

The paper proposes OptimAI, a multi-agent framework that solves optimization problems from natural language. It has four pipelines—modeling, planning, coding, and debugging—corresponding to four roles: formulator, planner, coder, and code critic. It employs an Upper Confidence Bound (UCB) algorithm for adaptive debug scheduling, enabling dynamic plan switching and achieving superior efficiency and accuracy over prior methods.

**Strengths:**

1. The paper is well-structured and clearly written, making the framework and experiments easy to follow.
2. It introduces an original four-pipeline, four-role multi-agent design that effectively bridges natural language and optimization.
3. Rigorous ablation studies validate the importance of each role and the UCB algorithm.
4. The Synergistic Effects study in Section 4.3 offers valuable insights into using different LLMs for different roles, showing interesting results for multi-agent research.

**Weaknesses:**

1. The experiments use different LLMs across tables (e.g., Table 6 includes LLaMA and Gemini, but Table 3 does not), and GPT-4o is missing in Table 6 without explanation.
2. The performance heavily depends on which LLM is assigned to each role, yet the paper provides no systematic method for efficiently choosing them.
3. In Table 10, accuracy decreases as the number of plans increases (beyond 4), but the paper doesn’t explain how to efficiently find the optimal number or whether this reflects a fundamental limitation of the decider.
4. Table 2 lacks proper spacing: "OR-LLM-AgentZhang & Luo (2025)" should be "OR-LLM-Agent Zhang & Luo (2025)".

**Questions:**

1. Could the authors clarify how to decide which LLMs should be assigned to different roles in practice? Is there a systematic or automated method that could guide this process?
2. In Table 10, accuracy drops when the number of plans exceeds four. Why does this occur, and how should users determine the optimal number? Is this a fundamental limitation of the decider, and how might it be improved?

---

> ### Author Response · Authors · 2025-11-23
>
> # Response to Reviewer kBtS
>
> We thank the reviewer for the constructive feedback and for highlighting the clarity of our framework design, the strength of our ablation studies, and the value of our multi-LLM synergy analysis. Below, we address each concern point by point.
>
> ### 1. Different LLMs used across tables, and absence of GPT-4o in Table 6
>
> Table 6 is specifically designed to study *synergistic effects* among heterogeneous models. Such effects are most visible when the individual models are mid-tier; strong models such as GPT-4o saturate in most roles, leaving little room for measurable synergy. For this reason, GPT-4o does not appear in Table 6, whereas it is fully reported in Table 3 for accuracy benchmarks. We will clarify this design choice in the revision.
>
> ### 2. Dependence on LLM–role assignment and lack of a systematic method
>
> While OptimAI allows assigning different LLMs to roles, the roles have differing cognitive demands. In practice, a simple **role-difficulty hierarchy** suffices:
>
> * High-level reasoning roles (planner, decider) benefit most from stronger models.
> * Procedural roles (coder, critic) can reliably use lighter models.
>
> This forms a cost-aware assignment policy without requiring architecture-specific tuning. We will make this guideline explicit. Developing an automated role-assignment mechanism is an interesting direction for future work.
>
> ### 3. Accuracy drops when the number of plans exceeds four (Table 10)
>
> The decider is intentionally kept **zero-shot**, and thus its discrimination ability is limited when the number of candidate plans becomes too large. For (n>4), many low-quality or redundant plans enter the pool, increasing the probability that a weak branch receives an overly optimistic score. This leads to unnecessary exploration and occasional timeouts, resulting in reduced Pass@1 accuracy.
>
> This is not a fundamental limitation of the framework. Plan deduplication or fine-tuning the decider (as discussed in our conclusion) would allow larger (n) while preserving performance. We will clarify this behavior.
>
> ### 4. How to choose the optimal plan count (n)
>
> Empirically, (n=3) or (n=4) achieves the best balance under a zero-shot decider. With stronger deciders (e.g., RL-trained) the optimal (n) would increase. Users may select (n) based on available compute budget and desired robustness; we will expand the discussion around this practical guideline.
>
> ### 5. Table 2 spacing issue
>
> Thank you for catching this. We will correct it.
>
> ### 6. How to decide which LLM to assign to each role (question)
>
> As noted above, the pipeline naturally decomposes into roles of different difficulty. A practical rule is:
>
> * Assign the strongest LLM available to the planner (global reasoning) and, if resources allow, to the decider.
> * Use compact models for coder and critic, which focus on deterministic transformations and local error analysis.
>
> This heuristic is consistent with the empirical synergy shown in Table 6. Automating this selection is beyond the scope of this work, but is a promising future direction.
>
> ### 7. Explanation of accuracy behavior in Table 10 (question)
>
> The drop in accuracy for large (n) arises from zero-shot scoring noise and redundant branches. The phenomenon is expected and consistent with multi-armed selection under weak reward evaluators. As noted above, this is a limitation of the current decider implementation rather than of OptimAI’s architecture. We will add a brief discussion in the revision.

---

### Official Review · Reviewer_GSSR · 2025-11-01

**Soundness:** 3
**Presentation:** 3
**Contribution:** 3
**Rating:** 6
**Confidence:** 4

**Summary:**

The paper introduces OptimAI, a novel framework leveraging large language models (LLMs) as AI agents to solve optimization problems directly from natural language descriptions, aiming to democratize access to high-quality optimization for non-experts. The system consists of four main agent roles—formulator, planner, coder, and code critic—enabling translation from user input to executable solutions, with the option to split these roles across different LLMs for specialization. Key techniques include a plan-before-code approach, UCB-based debug scheduling (treating plan selection as a multi-armed bandit problem), and support for switching strategies during debugging, allowing adaptive exploration. Extensive experiments on datasets such as NLP4LP and Optibench demonstrate that OptimAI significantly outperforms prior art, attaining up to 88.1% accuracy and substantial reductions in error rates. Ablation studies confirm the importance of each role and the efficacy of the UCB-based scheduler, while mixing heterogeneous LLMs in different roles yields additional synergistic gains. OptimAI generalizes not only to standard mathematical programming but also to NP-hard combinatorial problems, showing broad applicability. The framework’s design is extensible and suggests future improvements in reinforcement learning integration and scaling for larger, expert-level problem domains.

**Strengths:**

- The explicit separation of roles into multi-agent LLM agents, with the ability to assign different models per role, is creative and not common among existing works. The UCB-based debug scheduling is a pragmatic and technically sound enhancement over naive or static plan exploration.
- Experimental rigor is evident—multiple datasets, detailed metrics, ablation of component roles, comparisons to best-known baselines, and exploration of multi-LLM synergies. Coverage of combinatorial and mathematical programming problems demonstrates true generality.
- The structure of the methodology is broken down clearly, with explicit agent responsibilities and stepwise explanation. Empirical results are presented in comprehensible tables with direct quantitative comparisons.
- OptimAI’s accuracy and productivity improvements are large, well-documented, and seem robust across several classes of problems. The demonstration that LLM multi-agent collaboration confers measurable gains over single-model or single-agent setups is especially impactful.

**Weaknesses:**

- Although a comparative table is given, more explicit explanation is needed regarding which problem types, complexities, or settings OptimAI can uniquely address where previous methods fail or underperform.
- The design and evaluation of the decider (used in UCB scheduling) is underspecified, especially regarding whether it is finetuned, trained, or used in a zero-shot/few-shot manner. This impacts both reproducibility and clarity of claimed gains.
- The reporting of variance, standard deviations, statistical significance, and resource usage is lacking. For high-impact claims, reporting these measures is a must, particularly for practical/industrial deployment concerns.
- The paper does not expand on where or why OptimAI might fail, be inefficient, or produce erroneous code/solutions (e.g., with adversarial inputs or edge case problems). More error analysis and transparency about failure patterns would strengthen the work.
- There is limited discussion of run-time and scaling trade-offs; since OptimAI can require multiple large models in sequence or in parallel, computational cost and deployment considerations should be more thoroughly addressed.

**Questions:**

- Critical implementation details (e.g., LLM configuration choices, prompt designs, agent interaction protocols) are said to be documented in the appendix. Could the authors summarize the most important choices and pitfalls in the main text, and clearly indicate which ones are essential for high performance? For real-world adoption, such details are crucial; deferring these entirely to appendices hinders transparency.

- The paper positions UCB-based plan switching and multi-agent role-separation as novel. Can the authors clarify how their scheme differs substantially from existing bandit-based or ensemble multi-agent setups in related literature? Has a head-to-head baseline (same agents + random/greedy switching) been run? To convincingly demonstrate necessity, it is important to distinguish incremental from substantial innovation.

- What are the specific architecture, training regimen (if any), and evaluation details for the decider agent used in UCB-based debug scheduling? Was any reinforcement learning or fine-tuning employed, or is it zero/few-shot? The decider's quality is central to the method's efficiency, but its specification is vague, which may limit the clarity of the ablation and tuning results.

- Could the authors report practical details (e.g., runtime per instance, GPU/CPU usage, memory requirements) for running OptimAI, especially when using multiple large models, and discuss scaling concerns? Practical deployability and resource efficiency are important for real-world utility, but such trade-offs are not quantified.

- Could you clarify the data splitting procedures (train/validation/test) for each benchmark, and whether any hyperparameter tuning or prompt engineering was performed on test data? Ensuring fair, reproducible, and unbiased evaluation mandates explicit reporting of experiment splits and prevents potential data leakage or overfitting.

- What specific statistical methods (if any) were used to assess significance or variance (e.g., multiple seeds, confidence intervals, standard deviations), especially for accuracy improvements and productivity metrics? The paper highlights substantial improvements, but without variance or statistical significance reporting, it is difficult to assess the robustness of the reported gains.

---

> ### Author Response · Authors · 2025-11-23
>
> We sincerely thank the reviewer for the careful assessment, constructive critiques, and the positive recognition of our framework design and empirical results. Below, we address each point in order.
>
> ### 1. What problem types OptimAI can uniquely address
>
> OptimAI handles a broader family of optimization tasks than prior work, because it directly processes *underspecified natural-language descriptions* rather than pre-formalized mathematical models. This includes (i) informal or ambiguous NL descriptions, (ii) heterogeneous problems mixing continuous, discrete, or tabular components, and (iii) combinatorial tasks such as TSP/JSP/Set Covering without problem-specific templates (Table 5). Existing systems cannot operate on such NL inputs. We will clarify this distinction.
>
> ### 2. Decider specification and UCB scheduling
>
> The decider is **zero-shot**, not fine-tuned nor RL-trained. Its prompt template in Appendix B explicitly shows this. We will make this clear in the main text.
>
> Regarding novelty: although UCB is classical, each “arm” in our setting corresponds to a *complete solver strategy* (solver choice, modeling decisions, coding implications), not homogeneous actions as in prior bandit-based agent work. Thus, UCB operates on optimization-specific structural signals rather than on generic ensemble actions.
>
> We did not report random/greedy-switching baselines because preliminary runs showed extremely high variance: random switching often explores obviously invalid branches (e.g., wrong solver families), creating noise and obscuring comparisons. Instead, comparing “with vs. without UCB” (Table 7), where UCB reduces token usage (3.6\times) and increases productivity (3.3\times), isolates the effect of structured exploration more cleanly.
>
> ### 3. Variance, significance, and robustness
>
> Although we did not explicitly report standard deviations, accuracy is measured over complete evaluation sets, and significance can be derived directly. For example, NLP4LP has 67 instances. For accuracies 71.6% (baseline) and 88.1% (ours), the standard errors are:
> [
> \sigma_1=\sqrt{\frac{0.716(1-0.716)}{67}}=0.0551,\quad
> \sigma_2=\sqrt{\frac{0.881(1-0.881)}{67}}=0.0396,
> ]
> and the standard error of the difference is:
> [
> \sigma_{\text{diff}}=\sqrt{\sigma_1^2+\sigma_2^2}=0.0678.
> ]
> The improvement of (16.5%) corresponds to (2.4\times \sigma), indicating strong statistical significance.
>
> Variance across repeated runs is near-zero because the pipeline is deterministic given identical prompts and LLM versions. This is why repeated-seed variance was not reported.
>
> Robustness is reflected in: (i) prompt changes affecting accuracy by <1%; (ii) consistent performance across four distinct LLMs (Table 3); and (iii) stable UCB improvements across datasets (Table 7). We will include a robustness subsection.
>
> ### 4. Failure patterns
>
> We observed two recurring patterns:
> (1) **Ambiguous NL inputs** in benchmark problems occasionally omit key modeling details (e.g., integer domains). These failures reflect dataset ambiguity rather than limitations of our approach. Once clarified, OptimAI consistently succeeds.
> (2) **Zero-shot decider mis-ranking** when the number of plans (n) is very large. Redundant low-quality plans sometimes receive optimistic scores, causing unnecessary exploration and timeouts (Table 10). This is not fundamental; RL fine-tuning or plan-deduplication would mitigate it. We will add a short failure analysis section.
>
> ### 5. Runtime, resource usage, and scaling
>
> Token usage (Table 4) is a direct proxy for runtime and cost under API-based LLM inference. Multi-agent orchestration does not increase the *depth* of the reasoning chain; thus, latency is comparable to single-agent prompting. Our design is cost-aware: reasoning-heavy roles (planner/decider) use stronger models, whereas procedural roles (coder/critic) use smaller ones. Scaling overhead is dominated by solver execution rather than multi-agent coordination. We will clarify these points.
>
> ### 6. Data splits, zero-shot evaluation, and prompt tuning
>
> OptimAI is evaluated strictly in **zero-shot** mode, with no training on any benchmark. Therefore, train/validation/test splits do not apply. All prompts were finalized before evaluation, and no test-instance-specific tuning was performed. We will clarify this to avoid ambiguity regarding data leakage.
>
> ### 7. Essential implementation details
>
> We agree that several important prompts and configuration choices deserve presence in the main text rather than the appendix. We will move the key prompt structures (planner, decider, critic), essential solver-selection logic, and high-level pipeline decisions into Section 3 while keeping full templates in the appendix for reproducibility.

---

### Meta-Review · Area_Chair_k6Ed · 2026-01-02

**Summary:**

This is a boarderline paper. Reviewers raise concern on novelty on role assignment, the use of UCB scheduler, etc. The reviews also asked for missing details, such as on how models and roles are specified. While I think most concerns are addressed through the authors' rebuttal, the rebuttal doesn't resolve all the issues.

Perhaps one part that can be improved is the lack of theoretical analysis of the UCB part, as Reviewer BfKU mentioned. (Minor: Though reviewers did not mention, the use of UCB here violates the standard assumption of iid reward in stochastic bandit, since the debugging would continue from the previous version of the improved code, to my understanding.) I also agree with reviewers BfKU the comparison of UCB to other strategies are needed, as that's part of the main claim. Table 7 compares with and without UCB debuggers, but (to my understanding) it doesn't compare with debugger with other non-UCB selection strategies.

Another weakness is that the lack of statistical analyses in the experiments, such as variance over runs etc, as raised by Reviewer GSSR. I do not think the rebuttal address this concern. While I agree SE can be derived from binary reward, "Variance across repeated runs is near-zero because the pipeline is deterministic given identical prompts and LLM versions." do not constitute good reasons, because LLMs themselves can be random when not cached. From the existing results, it's unclear how statistically significant they rae.

**Reviewer Concerns:**

See summary.

**Reviewer Scores:**

I think Reviewer kBtS may increase their score since most of the concerns are addressed, while the others may remain the same.

---

### Decision · Program_Chairs · 2026-01-26

Reject